# Extracellular Vesicle-Based Therapeutics in Neurological Disorders

**DOI:** 10.3390/pharmaceutics14122652

**Published:** 2022-11-30

**Authors:** Yiwen Yuan, Jian Sun, Tongyao You, Weiwei Shen, Wenqing Xu, Qiang Dong, Mei Cui

**Affiliations:** 1Department of Neurology, Huashan Hospital, Fudan University, Shanghai 200031, China; 2State Key Laboratory of Medical Neurobiology and MOE Frontiers Center for Brain Science, Department of Neurology, Huashan Hospital, Fudan University, Shanghai 200031, China

**Keywords:** extracellular vesicles, neurological disorders, mesenchymal stem cells, drug delivery, therapeutics, neurodegenerative diseases

## Abstract

Neurological diseases remain some of the major causes of death and disability in the world. Few types of drugs and insufficient delivery across the blood–brain barrier limit the treatment of neurological disorders. The past two decades have seen the rapid development of extracellular vesicle-based therapeutics in many fields. As the physiological and pathophysiological roles of extracellular vesicles are recognized in neurological diseases, they have become promising therapeutics and targets for therapeutic interventions. Moreover, advanced nanomedicine technologies have explored the potential of extracellular vesicles as drug delivery systems in neurological diseases. In this review, we discussed the preclinical strategies for extracellular vesicle-based therapeutics in neurological disorders and the struggles involved in their clinical application.

## 1. Introduction

Neurological diseases such as stroke, neurodegenerative diseases, and neuromuscular diseases, are some of the leading causes of death and disability worldwide [1]. The therapeutics and drugs for treating neurological diseases are still limited. As the physiological and pathophysiological roles of extracellular vesicles (EVs) are becoming recognized, emerging evidence has demonstrated the therapeutic potential of EVs in treating neurological diseases [2,3]. EVs are heterogeneous, membrane-bound particles that are naturally released from most cells. They are generally classified into two types: endosome-origin “exosomes” and plasma-membrane-derived “ectosomes” (microvesicles) [4,5,6]. It is challenging to exactly distinguish exosomes and ectosomes because of their overlapping sizes and the similar markers between them [7]. Therefore, we used the generic term “EVs” to refer to both exosomes and ectosomes in this review.

EVs contain various types of cargos, including nucleic acids, proteins, and metabolites. Initial studies have focused on the regulatory and therapeutic effects of native EVs and their cargos. In the nervous system, EVs, as important mediators of intercellular communication in the brain, play a profound role in the maintenance and recovery of neural functions [8,9,10]. With the rapid development of nanotechnology, EVs have been modified for broader therapeutic capabilities, especially targeted drug delivery [11,12]. Some well-established synthetic nanoparticles such as liposomes, the first nano drug delivery systems, have received clinical approval and marketing authorization [13]. However, their therapeutic effects are limited because of their rapid clearance from the blood and their activation of the innate immune response [14]. Compared to synthetic drug delivery systems, EVs have emerged as a more complex and biocompatible form of liposomes with lower immunogenicity. The functional pharmacokinetic-related proteins of EVs contribute to their larger volume of biodistribution and higher retention in the circulation [15,16,17]. Furthermore, different molecules on the surface of EVs enable them to cross the blood–brain barrier (BBB), deliver their cargos, and evoke responses in the recipient cells [18,19,20,21,22,23]. These attributes of EVs exhibit their vast therapeutic potential in the treatments of central nervous system (CNS) disorders

The roles of EVs in the diagnosis, progression monitoring, and treatment of neurological diseases have been gradually recognized [24,25]. Inspired by emerging studies, here, we reviewed the preclinical evidence and strategies for EV-based therapeutics in the treatment of neurological diseases (Figure 1). Finally, we discussed issues to be considered for the future clinical applications of EVs.

## 2. Strategies for EV-Based Therapeutics in Neurological Diseases

### 2.1. EVs in Regenerative Therapies

The pathophysiological processes underlying most neurological diseases remain to be fully defined. Despite the delicate structures and functions of the nervous system, the spontaneous recovery of neurological function is commonly found in neurological diseases. Figuring out the underlying mechanisms and promoting the endogenous recovery process are key strategies for the development of therapies. Initial attempts were cell-based therapies such as the systematic administration of mesenchymal stromal cells (MSCs) in stroke models [26,27]. The safety of intravenous stem cell therapies has been approved in many clinical trials, but there are limited therapeutic effects in patients [28,29,30]. After different administrations, most stem cells are trapped in capillaries or peripheral organs such as the liver, spleen, and lungs [31]. The landmark paper by Gnecchi et al. proposed that paracrine factors account for the protective effects of MSCs [32]. Subsequently, emerging evidence has suggested that EVs secreted from MSCs contribute substantially to the beneficial effects of cell-based therapies [33]. Marbán drew interesting analogies between EV production and the process whereby bees manufacture honey [34]. He proposed that compared to fragile cells (like flowers), EVs (like honey), which contain the active ingredients of their sources, are more stable, efficacious, immune tolerant, and modifiable.

Regenerative progress is particularly important in post-stroke brain recovery, and the critical process is inducing neurogenesis and angiogenesis in lesions. Many studies have proven that MSC-EVs promote these agents in vivo and enhance functional recovery in stroke patients [35,36,37]. Although MSC-EVs promote neuronal survival and the process of neurogenesis in peri-infarct tissue, these studies have not shown significant effects in terms of reductions in the infarct volume. The failure to reduce the infarct volume was possibly attributed to the timing, because the time taken in MSC-EV delivery is usually longer than the 3–6 h window required for tissue plasminogen activator (t-PA) therapy, the only stroke treatment. Recent studies have indicated that neural-stem-cell (NSC)-derived EVs administered both within and outside the t-PA therapeutic window result in a significant reduction in infarct volume, exhibiting better therapeutic effects in stroke compared to MSC-EVs [38]. Modified NSC-EVs loaded with triiodothyronine, a critical factor for oligodendrocyte development, successfully target the demyelinated lesion in the brain and present an effective remyelination effect in experimental autoimmune encephalomyelitis (EAE), a classical multiple sclerosis (MS) animal model [39]. In addition, an injectable hyaluronic acid hydrogel has been developed to deliver NSC-EVs into the stroke brain, enhancing EV retention and sustaining the therapeutic effects [40]. However, considering the ethical questions and problematic logistics of acquiring fetal tissues, NSC is not an optimal choice as an EV producer. Induced NSCs from somatic-cells (iNSCs)-derived EVs display comparable therapeutic effects to NSC-EVs in post-stroke functional recovery without significant adverse effects [41]. The development of iNSCs can overcome some ethical questions about NSC-derived EVs, but the mass production of iNSC-EVs for clinical use is still a problem.

The function and proliferative capacity of stem cells decline over time. In regenerative therapies, the anti-senescence effect is another tool to promote endogenous recovery. For example, embryonic-stem-cell-derived EVs reverse hippocampal NSC senescence and the corresponding reduction in neurogenesis and cognitive impairment [42,43]. In addition, cellular senescence and stem cell exhaustion are two biological hallmarks of aging, which correlate with a susceptibility to neurodegenerative diseases and other aging-related diseases [44]. Recent studies have provided evidence for the potential roles of EVs in treatments for diseases related to aging and the hallmarks of aging. For instance, circulating EVs from young mice present a pro-longevity function [45,46]; EVs from fibroblasts of young human healthy donors ameliorate senescence in old recipient fibroblasts and mice, and they especially ameliorate oxidative stress—one of the classical pathophysiology features of senescence and aging [47]. Unfortunately, these therapeutical effects of EVs have not been deeply explored in most age-related neurological diseases.

### 2.2. EVs in Immune-Modulatory Therapies

Immune-cell-derived EVs have been favored as promising therapeutic agents for their immune-modulatory capabilities [48,49,50,51,52]. Microglia, the macrophage-like immune cells of the CNS, are involved in homeostasis and in the pathophysiology of CNS diseases [53]. EVs isolated from IL-4-treated microglia could inhibit glial scar formation, reduce brain atrophy volume, and promote white matter repair and functional recovery after ischemic stroke [54,55,56,57]. Increasing studies have shown that stem-cell-derived EVs present immune-modulatory capabilities. Embryonic-stem-cell-derived EVs significantly alleviate pathology and the long-term neurological deficits after ischemic stroke in a regulatory T-cell-dependent way [58]. The intrinsic anti-inflammatory activity of NSC-EVs also contributes to an improvement in pathology and behavioral outcomes in stroke [38,59]. The inhibition of leukocytes, especially polymorphonuclear neutrophil infiltration, may be the underlying mechanism of MSC-EV-induced ischemic neuroprotection [60]. MSC-EVs conjugated with LJM-3064 aptamer, a myelin-specific DNA aptamer demonstrating remyelination induction, on their surface produce immune-modulatory properties and promote remyelination in EAE [61]. Since MS is a typical autoimmune disorder in the CNS, researchers have also proposed using the programmed cell death protein-1/programmed death ligand-1 (PD-1/PD-L1) pathway to inhibit pathological immune responses and maintain self-tolerance in MS and EAE [62]. PD-L1-expressing dendritic cells demonstrate therapeutic effects in EAE, but there is still insufficient data for the immune-modulatory functions of PD-1/PD-L1-pathway-related EVs in MS [63]. Recently developed PD-L1-overexpressed MSC-EVs significantly suppress pathogenic immune responses via the PD1/PD-L1 pathway in two different autoimmune diseases: ulcerative colitis and psoriasis [64]. The application of these PD-L1-overexpressed EVs in MS may exhibit therapeutic effects and provide insights into EV-based immune-modulatory therapies in the treatment of autoimmune neurological disorders.

Naïve macrophage (Mϕ) EVs inherit the integrin lymphocyte-function-associated antigen 1 (LFA-1) from their parental cells and interact with the intercellular adhesion molecule 1 (ICAM-1) of BBB endothelial cells. In inflammation, the upregulation of ICAM-1 promotes the uptake of Mϕ EVs to the brain parenchyma [65]. Moreover, the brain-derived neurotrophic factor (BDNF), an important neuroprotective factor, loaded in these Mϕ EVs is stable and exhibits higher brain accumulation compared to BDNF alone [65]. Considering the common inflammatory response in neurological diseases and the properties of naïve Mϕ EVs, these EVs could be simple and efficient delivery vehicles of anti-inflammatory and neuroprotective drugs to brain lesions. However, similar to the distributions of other naïve EVs in a previous study, most naïve Mϕ EVs accumulate in peripheral organs such as the liver, spleen, gastrointestinal tract, and lungs, and less than 1% of administrated EVs successfully enter the brain [66]. We still need other methods to increase the brain-targeting efficiency of EVs and their cargos. In the next part, we focused on strategies for EV-based drug delivery to the brain, including advanced techniques for the efficient delivery of anti-inflammation drugs.

### 2.3. EVs for Targeted Drug Delivery

Most therapeutic agents are unable to reach lesions in effective concentrations by systemic administration because of biological barriers, especially the BBB, a major hindrance to drug treatments of CNS diseases. An alternative drug administration route is intranasal administration. The intranasal delivery route help drugs bypass the BBB and delivers them directly to the brain [67,68]. It has been proven to be an effective route for the delivery of drug-loaded EVs to the brain. For example, curcumin is a natural anti-inflammatory and antioxidant nutraceutical that has been widely investigated. Tumor-derived EVs with encapsulated curcumin are taken up by microglia via intranasal administration and have shown therapeutic effects in three independent inflammation-mediated disease models [69]. The intranasal administration of embryonic stem cell EVs loaded with curcumin not only reduces inflammation but also restores the neurovascular unit in ischemia-reperfusion-injured mice [70]. However, the nasal mucosa barrier limits the efficiency of intranasal therapy, as less than 1% of administrated drugs reach the brain [71].

Remarkable advancements in nanotechnology have yielded many brain delivery technologies that have achieved a sufficient delivery of drugs for neurological diseases [72]. Although the mechanisms by which EVs traverse the BBB and target lesions are still opaque, it is becoming increasingly difficult to ignore the role of EVs as drug delivery vehicles in the treatment of CNS diseases. However, as mentioned before, most naïve EVs tend to accumulate in peripheral organs. Surface modifications of EVs have been applied to increase their brain-targeting efficiency. For instance, cyclo(Arg-Gly-Asp-D-Tyr-Lys)-peptide (c(RGDyK))-conjugated EVs exhibit a high affinity to lesions of the ischemic brain after intravenous administration [73]. Curcumin loaded in these EVs exhibits a strong suppression of the inflammatory response in the lesion region of the ischemic brain [73]. In addition, the conjugation of rabies virus glycoprotein (RVG) to EVs displays a twofold greater accumulation in the brain [66]. As shown in the seminal work by Alvarez-Erviti et al., the conjugation of RVG to the EV surface significantly increased their brain-targeting efficiency [74]. Furthermore, exogeneous short interfering (si) RNA loaded in RVG-targeted dendritic cell EVs has been successfully delivered to the brain, resulting in a specific gene knockdown [74]. This landmark study not only demonstrated that EVs can be engineered into an efficient brain delivery system but also established an siRNA RVG-EV strategy for inhibiting the production of pathological proteins in neurodegenerative diseases.

Neurodegeneration is defined as the progress of loss or dysfunction of neurons, which impairs the properties of the CNS and previously established CNS functions such as mobility, coordination, memory, and learning [75]. One of the pathological hallmarks of neurodegenerative diseases is the aggregation of characteristic proteins. Representative instances are β-amyloid (Aβ) extracellular plaques and tau neurofibrillary tangles in Alzheimer’s disease (AD). Transactive response DNA-binding protein 43 (TDP-43) is a misfolded protein typical in amyotrophic lateral sclerosis (ALS) and frontotemporal dementia (FTD). Alpha-synuclein aggregates are the representative pathology in Parkinson’s disease (PD), dementia with Lewy bodies (DLB), and multiple system atrophy (MSA). Disease-specific pathological proteins and rare mutations of their encoding genes raise an interlinked hypothesis that a causal association exists between these proteins and neurodegenerative diseases [76].

One of the therapeutic strategies for neurodegenerative diseases is reducing the pathological protein burden in the brain. Alvarez-Erviti and colleagues first employed RVG-EVs loaded with exogenous BACE1 siRNA to inhibit the expression of BACE1 and Aβ in the cortex by systemic injection [74]. Subsequently, systemic siRNA RVG-EV treatment successfully reduced the expression of alpha-synuclein throughout the brain of S129D transgenic mice [77]. Likewise, RVG-EVs loaded with anti-alpha-synuclein short hairpin RNA (shRNA) minicircles decreased the aggregation of alpha-synuclein, rescued dopaminergic neurons, and improved clinical symptoms in a progressive mouse model of PD [78]. Indeed, some naïve EVs exhibited a capacity for reducing the pathological protein level. EVs secreted from adipose-derived MSC effectively ameliorated neurodegeneration and rescued cognitive impairment in APP/PS1 transgenic mice, a classic AD mice model [79]. Similarly, aFGF-stimulated astrocyte-derived EVs alleviated the brain Aβ burden and cognitive deficits in the APP/PS1 mice [80]. Adipose-derived EVs also reduce the aggregation of mutant superoxide dismutase 1 (SOD1), a pathologic hallmark of familial ALS [81]. Future studies may consider a combination of these naïve EVs and pathological-protein-targeted siRNA or shRNA to increase their therapeutic effects.

Due to the complicated aeotiology and pathogenesis of neurodegenerative diseases, apart from pathological-protein-targeted therapeutics, other EV-based therapeutic strategies have been developed. To start with, neurodegenerative diseases are commonly linked to neurotoxic substances such as reactive oxygen species (ROS), and hence the delivery of antioxidant drugs to the brain can be instrumental. EVs loaded with catalase effectively protected the substantia nigra pars compacta neurons from 6-OHDA-induced oxidative stress in a typical PD mouse model [82]. In addition, compared to free dopamine, EV-based dopamine delivery in PD models displays a more than fifteen-fold increase in brain distribution and a lower systemic toxicity [83]. Additionally, in a recent review, the authors summarized a therapeutic potential of EVs and small heat-shock proteins in neurodegenerative diseases [84]. They also hypothesized that a combination of them would be an effective therapeutic in the future by targeting the autophagy and apoptosis pathways, two major pathophysiological processes in neurogenerative diseases [84].

Engineered EVs fused with RVG have also been researched to carry many other compounds such as low-molecular-weight proteins, mRNA, and miRNA. For instance, the systemic administration of RVG-EVs loaded with nerve growth factor (NGF) protein along with NGF mRNA significantly increases the expression of NGF in the infarcted cortex, reduces inflammation, and promotes cell survival in stroke [85]. Considering that some proteins need post-translational modification or act in a paracrine way, the delivery of mRNA can promote the production of intracellular bioactive proteins in the targeted region [86]. EV-mediated protein and mRNA delivery provide a stable and efficient strategy to overcome obstacles such as the rapid extracellular degradation of naked proteins and mRNA. EV-based therapeutics may promote the development of mRNA therapeutics. Similarly, modified EVs with RVG fused to the exosomal protein lysosome-associated membrane glycoprotein 2b (Lamp2b) can deliver miR-124 to the infarct site and promote neurogenesis in stroke [87]. Among the various RNA delivery systems, EVs have been proven to be more efficient than other clinically approved synthetic systems [88]. As potential therapeutic targets or drugs in neurological diseases, the role of exosomal miRNAs has been comprehensively discussed in other reviews [89].

With recent advances in therapeutic nucleic acids, EV-based delivery has demonstrated great promise in nucleic acid therapies for genetic disorders [90]. Huntington’s disease (HD) and Duchenne muscular dystrophy (DMD) are two neurogenetic disorders that have been widely researched in this field.

The expansion of CAG repeats in the Huntingtin (HTT) gene leads to HD. The delivery of oligonucleotide directly targeting HTT mRNA and silencing the expression of pathological proteins is a promising strategy for HD treatment. EVs loaded with hydrophobically modified siRNAs display a significant bilateral distribution and silencing of Huntingtin mRNA in mice [91]. Recently, Zhang et al. designed an interesting genetic circuit that could reprogram hepatocytes to produce RVG-EVs loaded with mutant HTT siRNA [92]. These EVs reduced the levels of mutant HTT in the cortex and striatum and ameliorated behavioral deficits in three different mouse models of HD [92].

DMD is an X-linked recessive, progressive neuromuscular disease caused by the loss of dystrophin [93]. One of the most promising therapies for DMD is exon skipping, using antisense oligonucleotides (ASOs) to restore the function of dystrophin. Some ASOs have been tested in clinical trials, such as eteplirsen and drisapersen, based on phosphorodiamidate morpholino oligomer (PMO) and a 2′-O-methyl phosphorothioate (2OMePS), respectively [94]. Due to the unsatisfied functional benefits in large clinical trials, improving the delivery of ASOs by EVs may be an approach to promote the efficiency of exon-skipping therapy. The combination of PMO and peptide-modified EVs demonstrates an 18-fold increase in dystrophin expression in the muscle cells of mice compared to other non-exosomal components [95]. Similarly, anchoring myostatin propeptide, a natural inhibitor of myostatin, to the surface of EVs increased their stability and delivery efficiency and thus significantly promoted muscle regeneration and functional rescue in a DMD mouse model [96]. The systemic administration of EVs from different origins can rescue muscle function by stabilizing muscle membranes [97]. In addition, EVs secreted from muscle cells have been discovered for several years, and several studies have shown their effects in maintaining muscle homeostasis [98,99]. These muscle EVs may exhibit protective effects on muscle cells and slow the progress of neuromuscular diseases. However, far too little attention has been paid to these muscle EVs. Future studies are needed to explore the therapeutic effects of naïve or modified muscle EVs in treatments for neuromuscular disorders.

### 2.4. Inhibition of the Production of Pathological EVs

Other than their therapeutic roles, EVs are vital factors in the progression of neurological diseases [2,100,101]. For instance, EVs from B cells, rather than immunoglobulins or complement, cause apoptosis in neurons and oligodendrocytes in MS [102,103,104]. The contribution of EVs to the progression of neurodegenerative diseases has been well summarized [105,106,107,108,109]. The inhibition of EV release has emerged as a novel therapeutic strategy. There are two basic approaches currently adopted to develop pharmacological inhibitors of EVs: the inhibition of EV trafficking (e.g., calpeptin, manumycin A, and Y27632) and the inhibition of the lipid metabolism (e.g., pantethine, imipramine, and GW4869) [110]. For a comprehensive review of these inhibitors and their proposed mechanisms of action, readers may refer to the outstanding review written by Catalano and O’Driscoll [110]. In addition, some enzymes participating in EV secretion, such as peptidyl arginine deiminases, are preserved from microbes to mammals [111]. Thus, compounds that effectively inhibit the production of bacterial EVs may be potential inhibitors of EV release in mammals [111].

However, as pivotal agents of intercellular communication, some EVs transfer information and mediate physiological functions. Interference with the biogenesis of all EVs brings undesired side effects, and effective methods to inhibit specific pathological EVs are still lacking. The maturation of recombinant adeno-associated viruses (AAVs) provides cell-type-specific gene-delivery vehicles for neuroscience research [112]. Moreover, the silencing of genes such as GTPase, Rab27a, and Rab27b has been shown to significantly reduce the biogenesis and release of EVs [113]. Using cell-type-specific AAVs to silence EV genes in the parental cells of pathological EVs may increase the specificity of this therapeutic strategy.

## 3. Issues to Be Considered for Clinical Applications of EV-Based Therapeutics

EV-based therapeutics have attracted attention from the academic community and several pharmaceutical companies [114,115]. Clinical trials for EV-based therapeutics in humans have been approved, and the number of them is rapidly increasing [116,117,118,119]. We summarized all the registered clinical trials of EV-based therapeutics in Table 1. Among these registered trials, there are currently three registered clinical trials related to neurological problems: NCT04388982 (Alzheimer’s disease), NCT03384433 (cerebrovascular disorders), and NCT05490173 (neurodevelopmental disorders). MSCs were the source of EVs in all three trials, but the tissue sources of MSCs were different. Allogenic adipose MSCs were applied in NCT04388982 (Alzheimer’s disease), while allogenic bone marrow MSCs were applied in NCT03384433 (cerebrovascular disorders). Notably, NCT03384433 (cerebrovascular disorders) used allogenic MSCs transfected with miR-124, which may provide evidence for the effects of EV-based gene therapies in humans. In particular, the NCT05490173 (neurodevelopmental disorders) and NCT04388982 (Alzheimer’s disease) studies adopted intranasal administration, while the intravenous administration of EVs was applied in NCT03384433 (cerebrovascular disorders). We hope these representative clinical trials will provide exciting results for EV-based therapeutics in neurological diseases in humans.

As shown in Table 1, MSC-derived EVs are the most frequently used therapeutic EVs [121,122,123,124,125]. In the current bibliometric analysis of MSC-EV publications, 1595 articles and reviews about MSC-EV were published between 2009 and 2021; the annual publication reached as high as 555 publications in 2021 [126]. These data display a steep growth trend in MSC-EV studies in recent years. Given the rapid growth of preclinical and clinical MSC-EV publications, a systematic review has comprehensively analyzed the outcomes, methodologies, dosing, and possible mechanistic pathways of 206 animal studies using MSC-EVs as an intervention until 2020 [127]. Notably, a recent study has reported some interesting approaches that could enhance the therapeutic effects of MSC-EVs. For instance, appropriate preconditioning has been proven to enhance the therapeutic potential of MSC-EVs [128,129,130]. EVs of embryonic stem cells have an antisenescence activity on MSCs, which can be applied to maintain the EV production capacity of MSCs [131]. In addition, the pathology-specific homing of MSC-EVs is highly correlated with neuroinflammatory signals, suggesting that the distinct distribution pattern may be inflammatory-driven [132].

Indeed, as indicated in the explorations of MSC-EVs in laboratory and clinical studies, MSC-EV-based therapeutics have numerous problems. A major problem is their unstable therapeutic effects. For instance, studies offer contradictory findings about the effects of MSC-EVs on the angiogenesis process. MSC-EVs under hypoxic conditions promote angiogenesis, while MSC-EVs display anti-angiogenic effects in a pro-inflammatory microenvironment [133,134,135]. The unstable effects of native MSCs may be attributed to lots of reasons, ranging from isolation techniques, the cell source of MSCs, and distinct preconditioning. Though Gorgun and colleagues report that MSC-EVs display relatively minor changes in the microRNA landscape, they are less sensitive than soluble proteins in different microenvironments [136]. These results highlight the importance of standardized protocols in the preconditioning and elimination of soluble proteins.

The heterogeneity of EVs is another possible reason for their unstable therapeutic effects. Researchers have raised concerns about the different neuroprotective efficacy of MSC-EVs between individual preparations [60]. The latest review provides a definition of MSV-EVs and a general optimization strategy of the MSC manufacturing process for better and more homogeneous MSC-EVs [137,138]. On the other side, apart from avoiding the heterogeneity of EVs, understanding and taking advantage of the heterogeneity of EVs presents more opportunities for obtaining optimal therapeutic outcomes. For example, MSC-EVs consist of at least three types with different proteomes [139]. Recently, more exosome subpopulations have been found by asymmetric flow field-flow fractionation and high-speed ultracentrifugation [140,141]. These subpopulations within the classical subtypes of EVs mean that their therapeutic activities come from a collective effect of heterogeneous EV groups [142]. Although the biological significance and functions of these subpopulations have not been illustrated so far, some subpopulations may exhibit better therapeutic effects than other subpopulations. Thus, selecting the optimal subpopulations for treating different diseases will increase the consistency of their therapeutic effects and the accuracy of EV-based therapeutics.

Another major hindrance to clinical EV-based therapy is the production of sufficient EVs. Mendt et al. reported bioreactor-based progress in the large-scale generation of good-manufacturing-practice-grade (GMP-grade) MSC-EVs [143]. Plasma- and adipose-tissue-derived EVs have been proposed to be a cost-efficient alternative source of EVs compared to cell-culture-derived EVs [144]. Recently, a set of genetically encoded devices in EV-producer cells has been developed to enhance EV production and to specify the packaging and delivery of therapeutic mRNA in PD models [145]. In this study, the researchers designed an EV production booster, namely a tricistronic plasmid vector with a combined expression of STEAP3, syndecan-4 (SDC4), and a fragment of l-aspartate oxidase (NadB) at a fixed ratio, which produced a 15-fold to 40-fold increase in EV production in different cells. Co-transfecting the EV production booster, a potential RNA packaging device, a cytosolic delivery helper, mRNA, and RVG-Lamp2b into the EV-producer cells significantly increased the delivery of therapeutic mRNA into the target cells. Moreover, the engineered EV-producer cells implanted in living mice constitutively produced and delivered EVs loaded with therapeutic mRNAs, attenuating 6-OHDA-induced neuroinflammation in the PD mouse model. Thus, these genetically encoded devices can serve as efficient tools to produce a great deal of designer EVs and to enable the delivery of therapeutic mRNA in situ in vivo [145]. Grangier et al. critically compared the technological advances toward the mass production of EVs and clinical production platforms in detail [146]. The upscaling, isolation, storage, production, and standardization of them have been well discussed in other outstanding reviews [12,147,148,149,150,151].

The difficulty in the large-scale production of EVs for clinical application prompts the development of fully synthetic EV mimetic particles by bionanotechnology. EV mimetic vesicles can be a cheaper and more scalable nanosystem for encapsulating and delivering drugs. According to García-Manrique et al., the preparation of artificial EVs consists of two types: bioengineering cells as membrane fragment precursors (top-down methodologies) and mimicking the plasma membrane with artificial bilayers (bottom-up techniques) [152]. The mass production of EV mimetics with natural immunotolerance can be obtained by the top-down methodologies, but the encapsulation of cargo is passive and lacks selectivity. On the contrary, the bottom-up techniques provide specific cargo loading and high encapsulation efficiencies, while the production of EVs is still small. Therefore, the artificial EV field is still premature. Polyethylene-glycol (PEG)-induced membrane fusion between liposomes and EVs provides another strategy for this purpose [153]. The analysis of the characterization of EVs provides insights into the targeting and efficient delivery of therapeutic cargo, while the methodologies and advanced techniques in the liposome field may help to manufacture therapeutic EVs for specific clinical uses. Given the involvement of EV-associated lipids in the pathological process of neurological diseases, the modulation of EV lipidome represents a possible therapeutic strategy [107].

Off-target activity and dilution effects block the development and progress of EV-based therapy to some extent. The precisely controlled release of drugs is another challenge in EV-based therapeutics. Intelligent EVs engineered with responsive modules can react to specific internal and/or external stimulation to release cargo. These intelligent EVs have been systematically reviewed recently [154].

## 4. Conclusions

Despite ongoing challenges, extracellular vesicles have emerged as promising therapeutics and drug delivery systems in recent years. Emerging EV-based therapeutics in preclinical studies of neurological diseases have set the stage for their clinical use. In addition, the combination of EV-based therapeutics and some therapeutics that are in rapid development, such as mRNA therapeutics, may provide insight into treatments for neurological diseases. However, unstable therapeutic effects, difficulties in the large-scale production of EVs, and off-target activities still limit the clinical applications of EV-based therapeutics. Although many issues remain to be addressed, as a consequence of emerging standardized protocols and guidelines for EV isolation and storage, EV-based therapeutics are not far away from clinical application in neurological diseases.

## Figures and Tables

**Figure 1 pharmaceutics-14-02652-f001:**
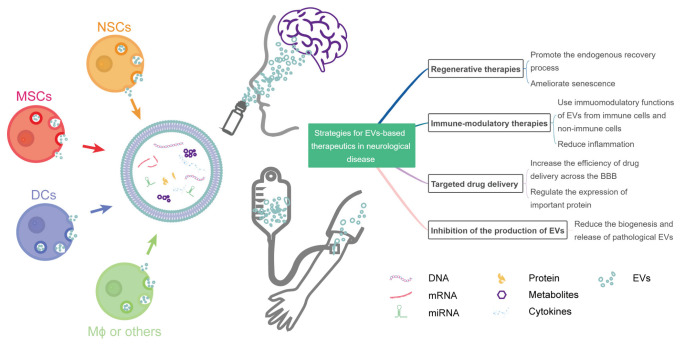
Strategies for EV-based therapeutics in neurological diseases. NSCs, neural stem cells; MSCs, mesenchymal stromal cells; DCs, dendritic cells; Mϕ, macrophage; BBB, blood–brain barrier; miRNA, microRNA; EVs, extracellular vesicles.

**Table 1 pharmaceutics-14-02652-t001:** EVs as therapeutics in registered trials.

Strategy	Study Identifier	Condition or Disease	Phase	Source of EVs	Native/Engineered	Status
Regenerative	NCT04388982	Alzheimer’s disease	I/II	MSC *	Native	Unknown
	NCT05490173	Premature birth Extreme prematurity Preterm intraventricular hemorrhage Hypoxia-ischemia, cerebral Neurodevelopmental disorders	NA *	MSC	Native	Not yet recruiting
	NCT04602104	Acute respiratory distress syndrome	I/II	MSC	Native	Recruiting
	NCT05402748	Fistula perianal	I/II	MSC	Native	Recruiting
	NCT04213248	Dry eye	I/II	MSC	Native	Recruiting
	NCT03437759	Macular holes	I	MSC	Native	Active, not recruiting
	NCT05060107	Osteoarthritis, knee	I	MSC	Native	Not yet recruiting
	NCT04270006	Periodontitis	I	Adipose stem cell	Native	Unknown
	NCT05475418	Wounds and injuries	NA	Adipose tissue	Native	Not yet recruiting
Regenerative and immune-modulatory	NCT04356300	Multiple organ failure	NA	MSC	Native	Not yet recruiting
	NCT04202770	Refractory depression Anxiety disorders Neurodegenerative diseases	NA	MSC	Native	Suspended
Immune-modulatory	NCT01159288	Non-small-cell lung cancer	II	dendritic cell	Native	Completed
	NCT04389385	Coronavirus infection Pneumonia	I	COVID-19 Specific T Cell	Native	Unknown
	NCT05191381	COVID-19 Critical illness Hypercytokinemia Lung fibrosis	NA	MSC	Native	Recruiting
	NCT04276987	COVID-19	I	MSC	Native	Completed [120]
	NCT04491240	COVID-19 SARS-CoV-2 pneumonia	I/II	MSC	Native	Completed
Drug delivery	NCT03384433	Cerebrovascular disorders	I/II	MSC	Engineered (loaded with miRNA-124)	Recruiting
	NCT04879810	Irritable bowel disease	NA	Ginger	Engineered (loaded with curcumin)	Recruiting
	NCT05043181	Familial hypercholesterolemia	I	MSC	Engineered (loaded with low-density lipoprotein receptor mRNA)	Not yet recruiting
	NCT03608631	KRAS NP_004976.2:p.G12D Metastatic pancreatic adenocarcinoma Pancreatic ductal adenocarcinoma Stage IV pancreatic cancer AJCC v8	I	MSC	Engineered (loaded with KRAS G12D siRNA)	Recruiting

* MSC, mesenchymal stem cell; NA, not applicable.

## Data Availability

Not applicable.

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
