# Peer review of "Extracellular Vesicle-Based Therapeutics in Neurological Disorders"

_pharmaceutics, 2022, doi:10.3390/pharmaceutics14122652_

Round 1

Reviewer 1 Report

This is a good review article describing the role of extracellular vesicle-based therapeutics in neurological disorders. However, the following are some suggestions that must be considered to make this review great.

1. One suggestion to the author is to have a general schematic cartoon and demonstrate ECV's role in different neurological disorders or delivery vehicles and put it in the introduction part. It would be easier for a reader to understand.

2. The author can put a cautionary note at the end that describes, why extracellular vesicle nanomedicine or advanced such kind of formulations did not result in any clinical benefits or the limitations associated with these vesicles for clinical therapy.

3. Line 84. What is the author's opinion about the lacking of infarct volume associated with MSC-EVs? The authors didn’t describe the reason for enhancing function recovery in stroke. If possible, Please add some lines from the updated literature. 

4.  Line 172: comprehensive can be changed into “comprehensively”.

Reviewer 2 Report

The review presented by Yiwen Yuan and colleagues addresses an interesting topic: the use of extracellular vesicles (EVs) for neurological disorders. A great number of relevant publications in the field are included, but I consider that authors need to improve the presentation, discussion, and organization of the cited works. Besides, there are numerous problems with the language use and style, and in consequence, the understanding of the manuscript gets more complicated. In the next lines I present my comments and suggestions, which I think that should be addressed in order to meet the standards for publication.

1. I consider that the title is not correctly presented. As in many previous works in Pharmaceutics and other journals, “Extracellular vesicle-based therapeutics” should be used. The same comment applies for the rest of “EVs based therapeutics” along the text, which should be “EV-based therapeutics”.

2. There are many other problems with the use of English, so an extensive editing is needed. Just a couple of examples. Line 27: “particles that naturally released from the cell”, something is missing. Line 41: “nano particles” should be a single word. Line 119: “A recent developed”. Line 150: “one of the pathology hallmarks”. Line 183: “EVs loading with…”.

3. Besides, I consider that authors need to use linkers. Many times, sentences about different topics are presented one after the other, and the reader cannot appropriately follow the meaning and implications of the presented information. For instance, the paragraph at lines 75-96.

4. In addition, I believe that authors had some problems with the references. References are not presented in order, some are missing… For example, after ref 43, the number 45 is presented (line 89), after the 67 comes the ref 80 (line 136), ref 97 is missing in the text, after the ref 139 comes the ref 145 (line 279). Moreover, I believe that a couple of sentences are not presented in the corresponding paragraph: Lines 78-80 “MSCs-derived EVs…” does not make sense there, and lines 113-115 “NSC-EVs not only improve…” does not make sense there.

5. Line 46: authors mention that “EVs are more biocompatible and immunologically inert”. Maybe authors wanted to say something different. EVs are not immunologically inert, as later mentioned in this manuscript (apart from the review by Robbins et al. in 2014, ref 55, I consider that the recent review of Buzas 2022 Nat Rev Immunol could be mentioned).

6. Line 88: Multiple sclerosis has not been presented in the text before.

7. Line 97: “In regenerative therapies, anti-senescence effect is not only another tool to promote the endogenous recovery, but also a possible therapeutic mechanism underlying the aging and aging-related diseases such as some neurodegenerative diseases.” I do not understand what authors wanted to say here. Anti-senescence is not “a possible therapeutic mechanism underlying the aging”. Could authors please rephase the sentence?

8. Lines 102-104: Reference number 51 is not about CNS disorders, diseases, or features. In any case, if authors consider that it is relevant for the present work, they should state why. Besides, in the “ameliorate senescence in old recipients and mice”. I consider that authors should specify that those EVs ameliorate senescence in old recipient fibroblasts and mice. And we can say that oxidative stress is ONE OF THE classical features of senescence and aging.

9. Lines 107-108: “microglia, a type of CNS immune cell”. This is a very vague sentence. I consider that authors should briefly mention the immune roles of microglia.

10. Line 118: Please explain the rational for the use of the LJM-3064 aptamer in MS models.

11. Lines 119-120: The paper cited in ref 63 did not work with MS models, and neither CNS models. I would like to ask the authors to explain what the applications of PD-L1 EVs are.

12. Lines 121-125: BDNF was not naturally present in macrophage EVs, researchers added it. I think that this point should be better explained. Moreover, I consider that authors should mention that most EVs did not reach the brain, and accumulated in other organs, and they should discuss the implications.

13. About curcumin: the works cited in refs 65-67 used different EVs (produced by different cell types) and different routes of administration. I believe that I would be interesting if authors comment about these points.

14. Lines 139-141: “Engineered EVs fused to rabies virus glycoprotein (RVG) have been researched to carry many compounds such as peptides, low-molecular-weight proteins, and bioactive compounds across the blood-brain barrier[80]”. The cited work mentions about RNAs in RVG-EVs, but I far as I saw, not about peptides and low-molecular-weight proteins. What are the works that did those experiments? In addition, I believe that the information presented in lines 167-180 would be better just after presenting the RVG-EVs. I those lines also, “EVs modified with RVG peptide are applied to deliver pathological proteins siRNA to the brain in neurodegenerative diseases”, this sentence does not make sense. EVs are applied to deliver siRNAs TO INHIBIT THE PRODUCTION of pathological proteins.

15. Lines 195-197: “EVs modified with RVG peptide are applied to deliver pathological proteins siRNA to the brain in neurodegenerative diseases[99]”. This is a review work, so it does not mean that the proposed hypothesis will work. Authors should specify this point.

16. Lines 198-200: “Apart from neurodegenerative diseases, with recent advances in therapeutic nucleic acids, the EVs-based delivery have demonstrated great promise in nucleic acids therapies of other neurological diseases[100].” As far as I can see, “other neurological diseases” are not commented in ref 100.

17. Lines 214-216: “EVs secreted from skeletal and cardiac muscle cells have been found for several years, but there is insufficient data for the physiologic and therapeutic functions of them in neuromuscular disorders[108,109]”. I find this sentence very vague. Could authors please further explain what they wanted to show and/or what are the implications?

18. Importantly, I think that section 2.4 should be further developed. I suggest that authors comment on the reviews by Catalano et al. 2020 (JEV) and Chen et al. 2022 (Frontiers Microbiol).

19. Regarding Figure 1, “lack of effective methods to inhibit specific pathological EVs” is not a strategy, but a limitation, so the figure should be adapted.

20. About the Table that authors prepared, in my opinion, the clinical trials of EVs in non-neurological disorders are not relevant for this manuscript. I consider that they should focus on neurological disorders and present the details of those trials. For instance, it is relevant to mention that both NCT04388982 and NCT03384433 will use allogenic MSCs. In addition, authors should try to include all the registered trials about therapeutic EVs in neurological problems. In a quick search, I found a clinical trial with MSC-derived EVs on long-term neurodevelopmental outcome in extremely low birth weight infants: NCT05490173.

21. Line 247: Ref number 122 is from 2020. I think that authors could comment that the field changes/advances fast.

22. Line 250: “Gold nanoparticle (GNP)-labeled MSC-EVs…” it is not relevant how they were labelled; the labelling is not responsible for the homing of EVs. I think that with point should be clarified.

23. Lines 267-269: “Besides, MSC-EVs are consisting of at least three types with different proteomes, and few studies compared the therapeutic potential of these subpopulations of EV[132,133]”. Authors need to explain in what aspects the 3 EV types are different, because in fact, each individual EV is different (the membrane composition and cargo is never identical). Besides, the references presented just afterward (134,135) are not about MSC-EVs. Then, that is the relevance of this studies? Why did authors mention them here? Please also clarify this point.

24. Lines 270-272: “The therapeutic activities may come from a heterogeneous EVs group, and we still do not figure out which fraction of EVs or which molecule of EVs really comes into play”. We still do not know the role of many component of EVs, but the role of some EV components have been described (as nicely commented by authors along the manuscript). Therefore, I consider that the sentence should be adapted.

25. Please mention and briefly explain that the work cited in ref 146 was carried out for Parkinson’s disease.

Reviewer 3 Report

The authors provided a well-documented overview of Ev's role application in the development of innovative therapeutics for neurological disorders. In addition to this, the review could represent a really interesting point of view in a field so dynamic and rich in potential future applications. The field of research focused on exosomes is in continuous evolution and even if the article is well written, the introduction section could be improved with a more general point of view about the application of EVs research as biomarkers sources for disease progression monitoring and stratification (PMID: 32932746). Figure 1 appears quite difficult to read, increase the font size. Conclusions could be improved by discussing the opportunities of the potential application of mRNA therapeutics.

I hope that my comments could be useful and I look forward to reading the revised version of the paper.

Good luck.

Round 2

Reviewer 2 Report

I appreciate the effort made by authors, the manuscript has notably improved. However, there are still some issues that I consider that need to be clarified before their work could be published. Here are my comments:

1.       An extensive English editing is needed. There are several problems along the text, and they impede a clear understanding of the presented information. Just some examples:

-          Lines 26-27. “EVs are defined as a branch of heterogeneous, membrane-bound particles that naturally released from cells”.

-          Lines 32-40. Please review the whole paragraph.

-          MSCs are abbreviated as “mesenchymal stem cells” in Figure 1, and as “mesenchymal stromal cells” in line 68. Please unify.

-          Line 71. “Many literatures…”

-          Line 76. “EVs” should be EV, without an “s”. The same problem is repeated in other section of the text and with other abbreviations.

-          Line 82. “Astrocyte derived” should be astrocyte-derived, with a “-“. Same problem in sections of the text and with other “-derived” cases.

-          Lines 87-94. The newly added text has several problems with the use of English. Besides, the meaning of “t-PA” or “tPA” was not introduced. 

2.       Line 97. There are several MS models, and the effect was shown in one of the (the EAE). Please specify this aspect.

3.       Lines 104-106. Authors moved the sentence, but I fell that it does not fit here either… There is no link to the previous sentences about NSC and iNSC-EVs.

4.       Lines 132-144. Authors included more information, but I think that it needs to be better organized and presented.

-          Now the EAE model is presented, but results of experiments in this MS model had been mentioned before (ref. 46 and 70). Therefore, I consider that the EAE model should be introduced the first time mentioned in the manuscript.

-          Authors did not mention the autoimmune diseases tested in ref. 71 (ulcerative colitis and psoriasis). I consider that they should be mentioned, as they are not neurological disorders.

-          Which are the concerns raised by researchers about the role of PD-1/PD-L1 in MS and EAE? Maybe authors wanted to say that other researchers had found interesting roles of PD-1/PD-L1 in MS and EAE?

5.       Lines 147-157. Authors introduced some of the requested information, but still some aspects are not clearly stated.

-          In ref. 74 most EVs also accumulated in other organs, and this is a relevant point.

-          The sentence “Considering the common inflammation response in neurological diseases, these instinct properties enable naïve MÏ• EVs a simple and efficient delivery vehicle of anti-inflammation and neuroprotective drugs to the brain lesions”. I think I understand what authors wanted to say, but the sentence is not correctly presented… Maybe: “Considering the common inflammatory response in neurological diseases and the properties of naïve MÏ• EVs, these EVs could be simple and efficient delivery vehicles of anti-inflammatory and neuroprotective drugs to brain lesions”. Is this what authors wanted to state?

6.       Lines 323-328: “Given the rapid growth of preclinical and clinical MSC-EV publications, a systematic review comprehensively analyzes the outcomes, methodologies, dosing, and possible mechanistic pathways of 206 MSC-EV publications until 2020[135]. In the current bibliometric analysis of MSC-EV publications, the annual publication reaches as high as 555 publications in 2021[136]. These data display a steep growth trend of MSC-EVs studies in recent years.” This comparison is not correct. The review of 2020 included the works of MSC-EVs as an intervention in animals (ref. 135). In contrast, ref. 136 included all the MSC-EV publications. Therefore, the numbers (206 until 2020 vs 555 in 2021) cannot be compared. Authors need to rephrase this part and make sure to clearly state what type of works were included in each of the publications.

7.       Lines 367-369: My suggestion about ref. 154 has partially been addressed. I think it would be very interesting for the readers if, apart from mentioning that it was for PD, authors could also briefly explain in the manuscript how the EV production booster works.
